# Simplification of Thermal Networks for Magnetic Components in Space Power Electronics

**David de la Hoz [1], Guillermo Salinas [1,\*], Vladimir Šviković [2] and Pedro Alou [1]**

[1]  Centro de Electrónica Industrial, Universidad Politécnica de Madrid, 28006 Madrid, Spain;
   david.delahoz.menendez@alumnos.upm.es (D.d.l.H.); pedro.alou@upm.es (P.A.)
[2]  Thales Alenia Space, 28760 Tres Cantos, Spain; vladimir.svikovic@thalesaleniaspace.com
\*  Correspondence: guillermo.salinas@upm.es

**Abstract:** The volume of magnetic components for space applications, directly related to the launch cost, and their performance are critically influenced by their capability to dissipate the internal electromagnetic power losses. This is the reason why accurate thermal models are required. Furthermore, these models need to be simple and versatile to allow a fast analysis of many different designs. In this study, a specific methodology to analyze inductors and transformers by thermal networks and a simplification based on the Thevenin's theorem leading to a simple equation are proposed. The specific details to address thermal modelling for space magnetic components are discussed. A generic 50 W Flyback transformer for space applications is analyzed in this study and experimental validation is provided, showing that the proposed method leads to a deviation in the temperature estimation between 1 °C and 5 °C, which is considered quite a good result from the literature review carried out.

**Keywords:** thermal network; power electronics; magnetic component; space application

---

## 1. Introduction

The spacecraft power systems are designed so that they are fully operational for 15 years [1], exposed to the extreme environmental conditions of space. Hence, thermal management of this equipment is crucial, since the main heat transfer mechanism at board level is conduction and there is no convection. Some regulations regarding the thermal limits for the on-board electronic equipment in satellites are defined by the MIL and ECSS standards [2–6]. Complying with these regulations ensures a safe operation from the thermal point of view, but might bring some thermal design challenges. These regulations are considered in this paper; as a result, the magnetic component analyzed in this paper must keep its hotspot temperature below 50 °C over the Printed Circuit Board (PCB) temperature. Further details will be explained along the paper.

Furthermore, the volume of a solid body influences its capability to dissipate heat: The higher the volume, the higher the heat exchange external surface and thus the better the dissipation of internal power losses to the ambient [7,8]. On the other hand, the mass and the volume of the power electronics equipment have a strong influence on the launch cost of a satellite [9] due to the large mass and volume percentage that it takes from the overall system [10]. It is important to mention that the magnetic components, such as inductors and transformers, constitute a considerably high percentage of the mass and volume budget of the power systems [11,12]. As a result, the design of magnetic components is one of the major concerns for power electronics designers.

Due to the previous reasons, using a proper thermal model to estimate the temperature rise of magnetic components is critical to ensure a safe thermal stability as well as to achieve a proper sizing of them. Since the direct solution of the heat equation is practically unfeasible for such as heterogeneous

and complex objects as magnetic components are, some thermal models can be found in the literature. These approaches can be grouped in three categories: Simplifications of the heat transfer equation, approaches using Finite Element Analysis tools or thermal networks, as explained next.

Starting with the simplifications of the heat transfer equation, a very simplified equation can be found in [13]. It represents the temperature rise of the magnetic component as a function of the power losses and volume, with an exponential coefficient that can be adjusted by prototype measurements. A modification was proposed in [14]. Both are very useful to have a first guess of the temperature rise due to the internal power losses, but their accuracy is very limited and can only be used for natural convection boundary conditions.

Another approach is the use of Finite Element Analysis (FEA) tools. They are considered as the most accurate option. An example can be found in [15]. However, the main disadvantages of this approach are that FEA tools are very computationally expensive, require certain expertise and they are not always available. The use of homogenization techniques is proposed in [16] to reduce the computational requirements without sacrificing the accuracy of the results, but the required time to solve each simulation is still considerably higher than solving the set of equations corresponding to a simple thermal network. Then, the aim of this paper is obtaining a fast and accurate thermal model valid for any input conditions, so that the temperature rise of the studied magnetic component can be obtained as accurately as FEA simulations while avoiding the corresponding time consumption and limitations.

Thermal networks are the last category and probably the most common. This approach is the most versatile, since a trade-off between complexity and accuracy can be achieved. One of the earliest attempts to use thermal networks for magnetic components can be found in [17], where a one-dimensional network is proposed. Some variations are proposed in [18,19], but still the results might considerably deviate from reality since in most cases the thermal distribution in a magnetic component is three dimensional. Other approaches like [20–23] consider the remaining cut planes, creating a 2D or 3D thermal network therefore achieving more accurate results. However, the first approach requires prototype measurements, which is not always possible and is time consuming; we tried to avoid this situation to create a model that can be use in a design stage without the need of measurements nor prototypes. The other approaches, on the other hand, treat the whole winding as a solid block of copper, incurring in certain deviation from reality. Another interesting approach can be found in [24], where a general cuboidal element is presented for 3D thermal modelling. The results of this approach are very accurate, but the cost is excessively increasing the granularity of the thermal network, hence increasing the complexity to formulate it.

Nevertheless, certain further considerations should be taken into account when designing a magnetic component for space applications. The main motivation of this paper is presenting a thermal model for space magnetic components that combines high accuracy, required to ensure a safe operation in the critical environment conditions of space electronics equipment, with the simplicity of the proposed simplification. This approach is meant to be applicable to any magnetic component, avoiding the excessive time consumption derived from the construction of prototypes or the use of FEA tools. In particular, a magnetic component designed for certain industry space application, depicted in Figure 1, will be analyzed in this study as a proof of concept. These specific concerns are addressed in this paper to achieve an accurate thermal network for this components in Section 2, which is the main contribution of this paper. One step further is carried out in this study: A generic simplification based on the application of the Thevenin's theorem is proposed, so that a very simplified thermal network can be achieved while ensuring accurate results, being the next contribution of this paper. Moreover, after this simplification, it is possible to translate this simplified network into an equation, which allows power electronic designers to analyze the thermal behaviour of a magnetic component in a simplified way. This constitutes the last contribution of this paper. The proposed methodology is explained along Section 2. Finally, the experimental validation of this methodology is provided in Section 3, where the generic 50 W Flyback tranformer for space applications from Figure 1 is tested by means of a suggested experimental setup.

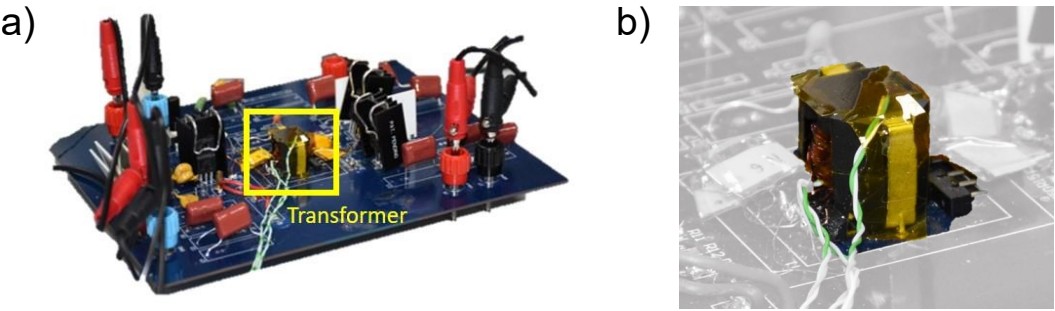

**Figure 1.** (**a**) Generic Flyback converter prototype for space applications analyzed in this study. (**b**) Detail of the modelled Flyback transformer.

## 2. Proposed Thermal Network Model

In this section, some specific concerns and assumptions regarding the thermal analysis of magnetic components for space applications are explained. Then, the proposed methodology to obtain an accurate thermal network for these components is explained step-by-step. Finally, a simplification of this network based on the Thevenin's theorem, as well as the equation that describes its behaviour, is proposed.

### 2.1. Governing Equations

From a thermal point of view, every analyzed domain of the magnetic component will be characterized by the general heat equation with internal heat generation [8]:

$$Q = -k\vec{\nabla}T \tag{1}$$

where $Q$ is the generated internal heat (if any), due to the power losses for the electromagnetic operation in this case, $k$ is the thermal conductivity of that domain and $\vec{\nabla}T$ is the temperature gradient of the domain. Nevertheless, it is not possible to obtain the analytical equation directly in objects with the geometric complexity of a transformer. On the other hand, the heterogeneity of its elements makes analysing it as a whole difficult, so it needs to be divided into simpler and more homogeneous geometries.

For this reason, the most common approach is using the electrical analogy and create a thermal network to discretize the problem. In this way, each thermal resistor on that network represents a portion of the analyzed object. The temperature rise from between the poles of that resistor is defined by the geometrical dimensions of that portion and its thermal conductivity. The basic expressions for these resistors can be found in the books about the fundamentals of heat transfer, like [7,8]. The expression of those resistors also depends on the used coordinates system (cartesian, cylindrical, spherical).

Nevertheless, those resistors only represent the temperature rise in objects without internal heat generation. In order to include the injection of the power losses in the thermal network, the linearization proposed by [17] will be used along this study. It consists in representing the corresponding volumetric fraction of the power losses generated in that portion of the object as a current source in the middle of two split thermal resistors. Further details regarding the thermal resistors and the power losses modelling will be explained along the next subsections.

On the other hand, the generated power losses must be calculated since they are the input of the thermal model. These power losses are provoked by the operation of the magnetic component in an electrical circuit. Many approaches can be found in the literature. In this study, the most common methods are used. The approach from [25] is used to calculate the winding losses, accounting for the high-frequency effects such as the 'skin' [26], 'proximity' [27] and 'gap' [28] effects, by means of a Finite Element Analysis electromagnetic solver. This is considered one of the most accurate approaches. In this case, the packages PExprt [29] and PEmag [30] from ANSYS Electronics Desktop 2019 R1 are used

to model the winding losses in the transformer.Regarding the core losses in the ferromagnetic core are calculated by means of the improved Generalized Steinmetz Equation [31], which shows a good consistency validated with measurements. It accounts for the 'hysteresis' [32,33] and 'eddy currents' [34] electromagnetic effects occurring in ferromagnetic materials under operation.

So far, the basic concepts regarding thermal modelling based on thermal networks and the power losses calculation in magnetic components that are used in this study are defined.

### 2.2. Boundary Conditions and Assumptions

Most magnetic components used in power electronics for space applications are mounted on a Printed Circuit Board (PCB), which are located within the metallic structure of a power electronics unit, such as a Power Control and Distribution Unit (PCDU). The temperature of these metallic structures is regulated by the Thermal Control subsystem, but every device mounted inside them mainly dissipate their generated power losses mainly via conduction, making the dissipation very challenging.

To analyze magnetic components for on-board power electronics equipment in spacecrafts, some considerations and assumptions can be applied to simplify the analysis without compromising the accuracy of the results:

- Heat transfer by radiation can be neglected at device level, since the temperature of nearby objects and the enclosure is close to the magnetic component's temperature. Furthermore, these electronic devices have operating temperatures lower than 120 °C, so the contribution of radiated heat transfer is much lower than the conductive heat transfer [7].
- There is no convective heat transfer since there is no air flow nor any surrounding fluid, due to the space environment conditions [1].
- The produced power losses caused by the electromagnetic operation of the magnetic component are dissipated via conduction through the Printed Circuit Board (PCB) in which it is mounted. This PCB is usually attached to the structure that, at the same time, is connected to the heat extraction system to deep space [35]. Hence, the PCB located at the bottom of the magnetic component can be considered as a temperature reference point or, in other words, as a constant temperature boundary condition [8].
- Power losses are assumed as concentrated in the corresponding thermal nodes from the thermal network and represent the corresponding volume fraction of the losses generated in that portion of the object, as current sources [17].
- Every magnetic component has at least one symmetry axis. Taking advantage of this symmetry considerably simplifies the thermal network, since half of the thermal resistors can be removed [8].

As a consequence of the previous statements, the external surfaces of the magnetic component are considered adiabatic, as well as its symmetry [8]. Its bottom surface is set at constant temperature and equal to the PCB temperature due to the nature of the system. This temperature is considered as the reference (or ambient temperature in a typical heat transfer problem), so that any other temperature rise is represented over that PCB temperature. These boundary conditions are marked in the cross-section representation of the analyzed transformer in Figure 2.

Regarding the thermal properties of the materials used for the magnetic in this study, a summary is presented in Table 1.

**Table 1.** Thermal conductivity of simulated materials, taken from [36,37].

| | k $\left[\frac{W}{m*K}\right]$ | | k $\left[\frac{W}{m*K}\right]$ |
|---|---|---|---|
| Ferrite | 4 | Pin connectors | 113 |
| Copper | 380 | Transformer bobbin | 0.2 |
| FR4 (PCB) | 0.35 | Insulation | 0.2 |
| Still air | 0.003 | | |

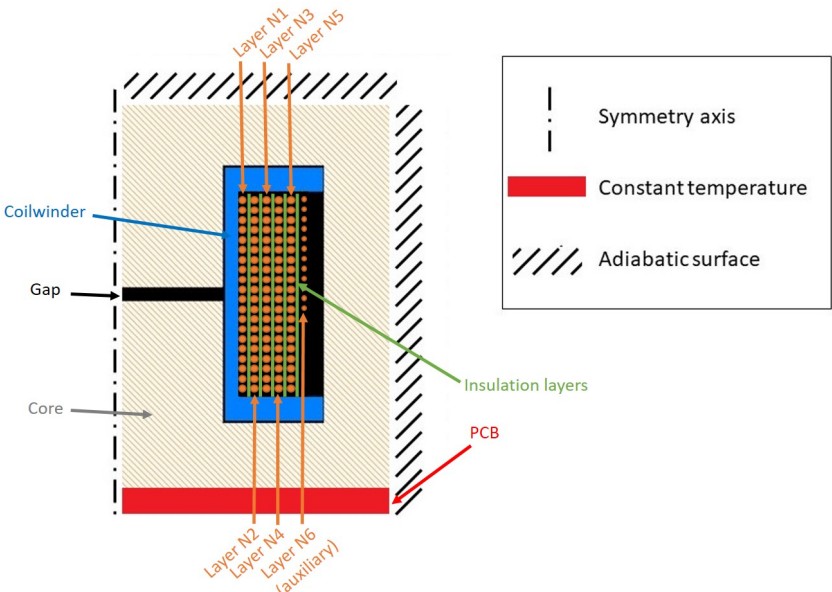

**Figure 2.** Cross-section of the analyzed transformer (Figure 1b) with the corresponding boundary conditions for the thermal analysis.

*2.3. Detailed Thermal Network*

Due to the complex geometry of high frequency transformers the first stage in the process to obtain the detailed thermal network consisted in dividing the geometry in regular domains which can be analyzed with analytical equations. The chosen domains for achieving that simplification were the windings and the core. Each winding was modelled as a torus and the core was divided in a set of connected prisms. On the other hand, to be able to assemble the complete thermal network a boundary condition of having the same temperature in the nodes of each of the subnetworks that will be connected is required.

This model assumes that the section of the transformer that corresponds to the windings has radial symmetry around the center axis and that the core has axial symmetry.

2.3.1. Network Linearization

In a transformer, power losses are produced in both the windings and the core. This means that the temperature field in those elements is not linear and, consequently, it is not possible to translate it into a resistor network without linearizing it. However, to obtain a precise thermal network, that linearization required different aproaches in both elements as a consequence of the obtained temperature profiles in both elements due to their dissimilar geometry and thermal conductivity.

The high thermal conductivity of copper and the small section of each conductor, allowed to consider the temperature in each section of the windings contant, as the major temperature increment is produced in the isolation layer that wraps the conductor due to the smaller thermal conductivity of the isolation material.

Contrarily, the lower thermal conductivity of ferrite causes a highly non-linear profile when core losses represent a significant part of the total heat produced in the transformer. Therefore, this component was linearized by meshing it with a similar process to the one used in a 1D FEA simulation, thus approximating its parabolic temperature profile by a piecewise linearized one. This allowed to control the error caused by the linearization by adjusting the number of elements used.

The process to obtain the thermal network associated to the winding and to the core of the magnetic component is described next.

2.3.2. Windings Thermal Network

As it was stated before, each winding turn can be modelled as a torus in which heat is just transmitted radially. Consequently, the obtained thermal profile in each section is the same that the one obtained in a cylinder in which heat is just transmitted radially, as expressed in Equation (2).

$$\frac{R_\rho}{L} = \frac{\ln\left(\frac{r_2}{r_1}\right)}{2\pi * k} \tag{2}$$

where $\frac{R_\rho}{L}$ is the radial thermal resistance of the hollow cylinder over length, $r_2$ is the outer radius, $r_1$ is the inner radius and $k$ is the thermal conductivity of the insulator that wraps the conductor.

Integrating Equation (2) along the torus the equivalent thermal resistor of each layer can be obtained as Equation (3).

$$R_{\rho,torus} = \frac{\ln\left(\frac{\frac{\phi_{section}}{2}+thickness}{\frac{\phi_{section}}{2}}\right)}{2\pi * (\pi * \phi_{torus}) * k} \tag{3}$$

where $R_{\rho,torus}$ is the thermal resistance of a hollow torus, $\phi_{section}$ is the inner diameter, *thickness* is the thickness of the hollow torus, $\phi_{torus}$ is the main diameter of the torus and $k$ is the thermal conductivity.

Nevertheless, each winding transmits heat in different directions, so the equivalent thermal resistor in each of them is required to obtain the complete thermal network, Equation (4).

$$R_{contact}\left(\varphi\right) = \frac{360°}{\varphi} * R_{\rho,torus} \tag{4}$$

where $R_{contact}$ is the thermal resistance of the contact surface of a toroidal insulated conductor and $\varphi$ is the contact angle.

Particularizing Equation (4) for a contact angle of 90 degrees, the equivalent thermal resistor of the insulation layer in each direction is 4 times the one of the complete layer, as shown in Equation (5).

$$R_{contact}\left(90°\right) = 4 * R_{\rho,torus} \tag{5}$$

This equivalent resistor can be used to obtain the thermal resistance between interleaved windings. Assuming that both windings have the same number of turns and conductors, the equivalent thermal resistance between interleaved windings can be obtained with Equation (6).

$$R_{interleaved\ winding} = \frac{2 * R_{contact}\left(90°\right)}{n_{turns} * n_{parallel\ conductors} * n_{windings} - 1} \tag{6}$$

where $R_{interleaved\ winding}$ is the thermal resistance between interleaved windings, $R_{contact}$ is the contact thermal resistance of each turn, $n_{turns}$ is the number of turns of each winding, $n_{parallel\ conductors}$ is the number of parallel conductors in each winding and $n_{windings}$ is the number of windings in the layer.

On the other hand, the total resistance between a layer of windings and the next layer of the transformer is obtained as the parallel of the contact thermal resistor of each turn, as Equation (7).

$$R_{winding\ layer} = \frac{R_{contact}\left(90°\right)}{n_{turns} * n_{parallel\ conductors} * n_{windings}} \tag{7}$$

where $R_{winding\ layer}$ is the radial thermal resistance of a winding layer.

Finally, the thermal resistance between each layer of conductors and the top or bottom surface of the transformer bobbin corresponds to a single equivalent contact resistor Equation (8).

$$R_{winding,\ transformer\ bobbin} = R_{contact}\left(90°\right) \tag{8}$$

where $R_{winding,\ transformer\ bobbin}$ is the thermal resistance between a winding and the transformer bobbin.

Once the thermal model of each conductor layer has been obtained, the equivalent resistance of insulating elements between layers is required to complete the thermal network. These layers can be modelled as a hollow cylinder in which heat can just be transferred radially due to its low thickness. Therefore, their equivalent resistance can be obtained by integrating Equation (2) along its height, which is expressed as Equation (9).

$$R_{insulating\ layer} = \frac{\ln\left(\frac{\frac{\phi}{2}+thickness}{\frac{\phi}{2}}\right)}{2\pi * height * k} \tag{9}$$

where $R_{insulating\ layer}$ is the radial thermal resistance of a cylindrical shaped insulator, $\phi$ is the inner diameter of the insulator, *thickness* is its thickness, *height* is the height of the layer and $k$ is the thermal conductivity of the insulator.

The other insulating layer that needs to be considered is the transformer bobbin. However, as this layer is also in contact with the core, it needs to fulfill the boundary conditions to allow the connection of both thermal networks. Thus, its model should have the same number of nodes in its external surface as the core network and a single node per winding layer in its internal one. To achieve this, the thickness of each element of the uper part will be chosen to have the same value as the diameter of the conductors used in the transformer. This ensures a one-to-one correspondence between both thermal networks, allowing to model the top and bottom surfaces of the transformer bobbin as a set of disks in which heat is just transferred axially. Consequently, the equivalent resistor that models each of those disks can be obtained with Equation (10), whereas its internal leg can be modelled with the same equation used for insulating layers between winding layers, Equation (9).

$$R_{disk} = \frac{Height}{k * \frac{\pi}{4} * \left(\phi_{max}^2 - \phi_{min}^2\right)} \tag{10}$$

where $R_{disk}$ is the axial thermal resistance of a disk, *Height* is its height, $\phi_{max}$ and $\phi_{min}$ are the inner and outer diameters of the disc and $k$ is the thermal conductivity of the material.

After all the equivalent models of the elements that compose the winding thermal network have been obtained, it is possible to assemble the complete network for the windings obtained for this application, which is shown in Figure 3. The values of the components shown on that schematic can be found in Table 2.

**Table 2.** Resistor values of the windings thermal network for the analyzed transformer (Section 3.2).

| Identifier | Value $\left(\frac{K}{W}\right)$ | Equation | Identifier | Value $\left(\frac{K}{W}\right)$ | Equation |
|---|---|---|---|---|---|
| $R_{contact}$ | 7.32 | (5) | $R_{n1,side}$ | 0.15 | (7) |
| $R_{n2,side}$ | 0.15 | (7) | $R_{n3,side}$ | 0.31 | (7) |
| $R_{n4,side}$ | 0.20 | (7) | $R_{n5,side}$ | 0.31 | (7) |
| $R_{n6,side}$ | 0.15 | (7) | $R_{n7,side}$ | 0.15 | (7) |
| $R_{n1}$ | 0.16 | (6) | $R_{n3}$ | 0.33 | (6) |
| $R_{n5}$ | 0.33 | (6) | $R_{n7}$ | 0.16 | (6) |
| $R_{inner\ kapton}$ | 0.55 | (9) | $R_{outer\ kapton}$ | 0.38 | (9) |
| $R_{TB,down\ 1}$ | 610.61 | (10) | $R_{TB,down\ 2}$ | 577.40 | (10) |
| $R_{TB,down\ 3}$ | 547.62 | (10) | $R_{TB,down\ 4}$ | 520.76 | (10) |
| $R_{TB,down\ 5}$ | 496.41 | (10) | $R_{TB,down\ 6}$ | 474.23 | (10) |
| $R_{TB,down\ 7}$ | 453.96 | (10) | $R_{TB,up\ 1}$ | 360.81 | (10) |
| $R_{TB,up\ 2}$ | 341.19 | (10) | $R_{TB,up\ 3}$ | 323.59 | (10) |
| $R_{TB,up\ 4}$ | 307.72 | (10) | $R_{TB,up\ 5}$ | 293.33 | (10) |
| $R_{TB,up\ 6}$ | 280.23 | (10) | $R_{TB,up\ 7}$ | 268.25 | (10) |
| $R_{TB,leg}$ | 156.62 | (10) | | | |

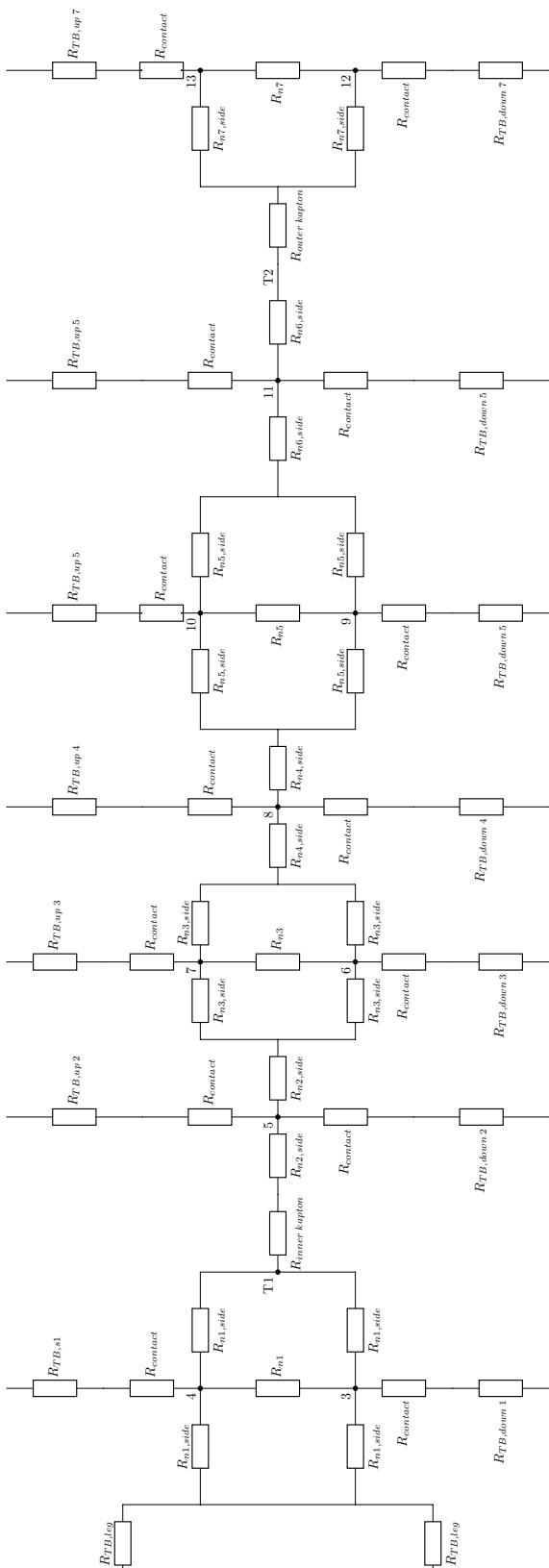

**Figure 3.** Equivalent thermal network of the windings of the analyzed transformer (Section 3.2).

Additionally, the thermal resistance between winding layers connected in series and the one between the windings and the PCB need to be modelled. Both the pins and the conductors can be modelled as a cylinder in which heat is transferred axially Equation (11).

$$R_{cylinder} = \frac{Length}{A_{cylinder} * k} \tag{11}$$

where $R_{cylinder}$ is the axial thermal resistance of a cylinder, *Length* is its length, $A_{cylinder}$ is the area of its section and $k$ is the thermal conductivity of its material.

By obtaining parallel or series equivalent of those elements, the thermal network that describes their behaviour can be obtained, shown in Figure 4.

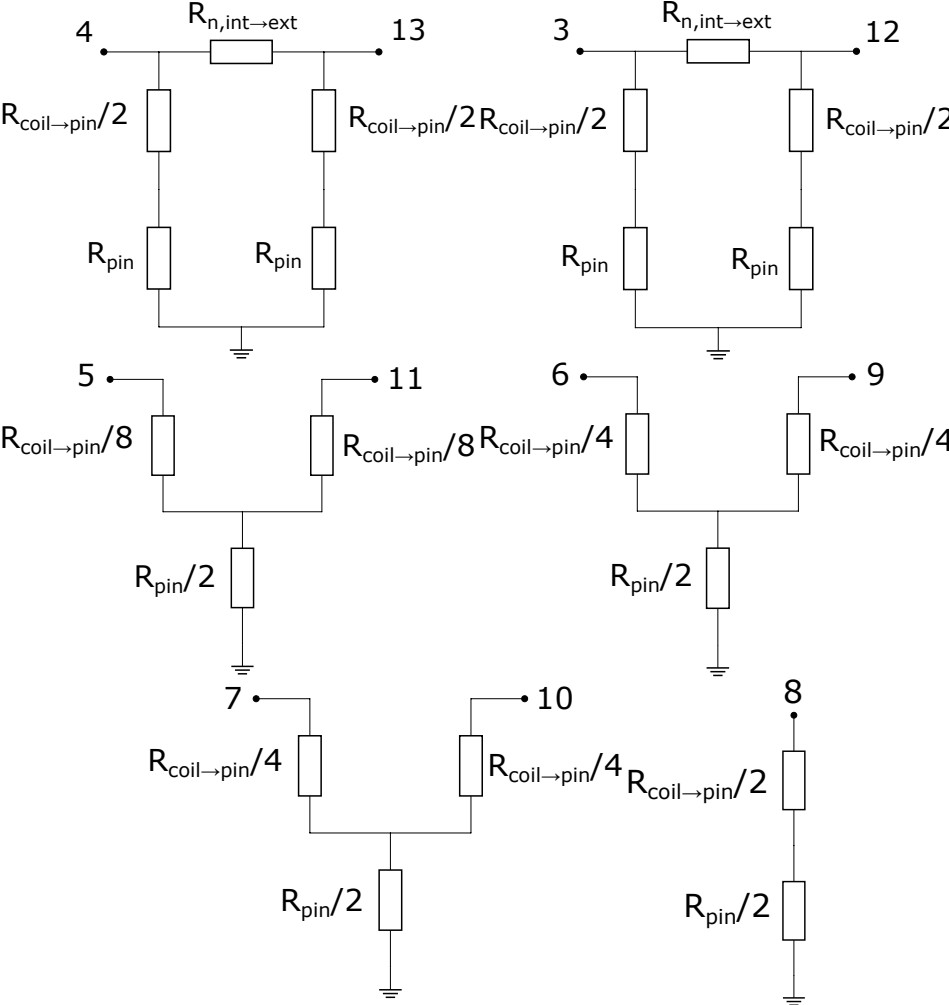

**Figure 4.** Equivalent thermal network of connection between windings and Printed Circuit Board (PCB).

Where the values of its resistors can be found in Table 3.

**Table 3.** Resistor values of the windings thermal network for the analyzed transformer (Section 3.2).

| Identifier | Value $\left(\frac{K}{W}\right)$ | Equation | Identifier | Value $\left(\frac{K}{W}\right)$ | Equation |
|---|---|---|---|---|---|
| $R_{n,int \to ext}$ | 44.74 | (11) | $R_{coil \to pin}$ | 535.86 | (11) |
| $R_{pin}$ | 91.66 | (11) | | | |

Up to this point, the process to obtain the thermal network corresponding to the winding is described.

### 2.3.3. Core Thermal Network

In order to model the thermal behaviour of the core, it needs to be divided into simpler geometries that are easier to model analytically due to the complexity of its geometry. Therefore, a RM/I core can be divided in four parts [38].

The internal leg can be described as a set of cylinders connected in series in which heat is transferred axially, so their equivalent resistance can be obtained with Equation (11). The thermal conductivity of each of those cylinders corresponds to the one of the ferrite or the material used in the gap (still air in this case; see Table 1). The same happens with the external leg, in which each element has a disk sector shape. Consequently, the general equation of heat transfer solved for axial transference, Equation (12) must be used in this leg, as well as in the mounting clips.

$$R_{axial} = \frac{\Delta h}{A * k} \tag{12}$$

Diversely, the upper part of the core can be modelled as a set of prisms connected in series with a thickness equal to a conductor diameter, Equation (13), so it is possible to assemble this network with the transformer bobbin as described in the previous section.

$$R_{core,upper\ element} = \frac{\phi_{conductor}}{2 * A * k} \tag{13}$$

where $R_{core,upper\ element}$ is the thermal resistance of an element of the upper part of the core, $A$ is the area of its section and $k$ is the thermal conductivity of the ferrite.

However, the equivalent resistance obtained with this method might be imprecise near the central leg, as the diameter of the coilwinder disk to which it will be connected can be smaller than the width of the upper part of the core. Thus, the resistance of this element are more similar to the one of a disk in which heat is transferred radially, Equation (14).

$$R_{core,upper\ element,corrected} = \begin{cases} \frac{\phi_{conductor}}{2*A*k} & \text{if } \phi_{TB\ disk} > core\ upper\ part\ width \\ \frac{\ln\left(\frac{\frac{\phi_{TB\ disk}}{2}+\phi_{conductor}}{\frac{\phi_{TB\ disk}}{2}}\right)}{2\pi*height*k} & \text{if } \phi_{TB\ disk} < core\ upper\ part\ width \end{cases} \tag{14}$$

where $R_{core,upper\ element,corrected}$ is the thermal resistance of an element of the upper part of the core, $A$ is the area of its section, $\phi_{transformer\ bobbin\ section}$ is the diameter of the the transformer bobbin ring to which the core element will be connected, $\phi_{conductor}$ is the diameter of a conductor used in the layer connected to that transformer bobbin ring, *height* is the height of the element of the upper part of the core and $k$ is the thermal conductivity of the ferrite.

Lastly, the lower part of the core is considered to have the same temperature as the PCB so it is connected to ground, which is the temperature of the PCB considered as the reference temperature. As a consequence, there is no need to model it as a thermal resistance.

As a result of these modelling processes, the thermal network obtained for the core of the transformer is shown in Figure 5, whose values of the resistors correspond to the ones found in Table 4.

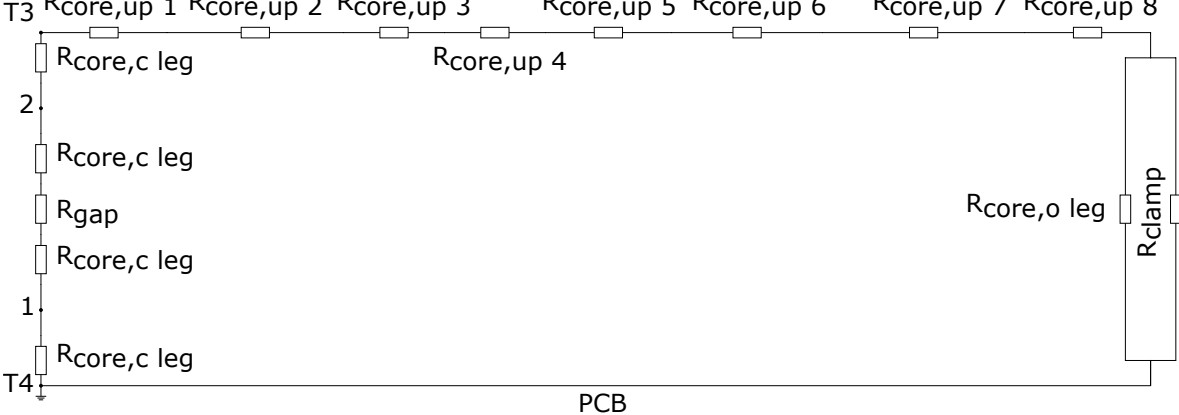

**Figure 5.** Equivalent thermal network of the core of the analyzed transformer (Section 3.2).

**Table 4.** Resistor values of the core thermal network of the analyzed transformer (Section 3.2).

| Identifier | Value $\left(\frac{K}{W}\right)$ | Equation | Identifier | Value $\left(\frac{K}{W}\right)$ | Equation |
|---|---|---|---|---|---|
| $R_{core,c\,leg}$ | 38.56 | (11) | $R_{gap}$ | 6.77 | (11) |
| $R_{core,up\,1}$ | 9.43 | (14) | $R_{core,up\,2}$ | 1.37 | (14) |
| $R_{core,up\,3}$ | 1.37 | (14) | $R_{core,up\,4}$ | 1.37 | (14) |
| $R_{core,up\,5}$ | 1.37 | (14) | $R_{core,up\,6}$ | 1.37 | (14) |
| $R_{core,up\,7}$ | 1.37 | (14) | $R_{core,up\,8}$ | 21.16 | (14) |
| $R_{core,o\,leg}$ | 510.20 | (12) | $R_{clamp}$ | 36.89 | (12) |

### 2.3.4. Heat Injection in the Thermal Network

Once the thermal networks for both the winding and the core of the transformer are developed, the power losses generated dissipated as heat need to be modelled. The heat produced in each element of the transformer can be modelled as a current source connected between ground and the node that represent that element.

Accordingly, the heat produced in each winding must be modelled as a set of current sources connected to the node that represents each layer of the winding, being possible to obtain the value of each source with Equation (15).

$$I_{winding\,layer} = \frac{N_{layer}}{N_{winding}} * Q_{winding}$$

(15)

where $I_{winding\,layer}$ is the current that needs to be inyected in that winding layer to simulate the effect of the produced heat in that winding, $N_{layer}$ is the total amount of turns of that winding in the layer including the ones of each parallel conductor, $N_{winding}$ is the total number of turns of the winding including the ones of all the parallel conductors used in it and $Q_{winding}$ is the heat produced in that winding (calculated as explained in Section 2.1).

With regard to the power losses produced in the core (obtained as explained in Section 2.1), heat losses can be modelled as a set of current sources connected to each of the nodes of its thermal network, being the value of each current source described by Equation (16).

$$I_{core\,element} = \frac{V_{element}}{V_{core}} * Q_{core}$$

(16)

where $I_{core\,element}$ is the current that needs to be injected in the node that represents that node to model its loses, $V_{element}$ is the volume of the element, $V_{core}$ is the total volume of the core and $Q_{core}$ is the heat produced in the core.

Nevertheless, if winding losses prevail over the ones produced in the core, it is possible to simplify the model by assuming that all losses are produced in the central leg without affecting the final result. Thus, Equation (16) can be simplified as Equation (17).

$$\begin{cases} I_{core\,element,central\,leg} = \frac{V_{element}}{V_{central\,leg}} * Q_{core} \\ I_{core\,element,outside\,central\,leg} = 0 \end{cases}$$

(17)

where $I_{core\,element}$ is the current that needs to be injected in the node that represents that node to model its loses, $V_{element}$ is the volume of the element, $V_{central\,leg}$ is the volume of the central leg of the core and $Q_{core}$ is the heat produced in the core.

As a consequence heat injection can be modelled with the circuit shown in Figure 6.

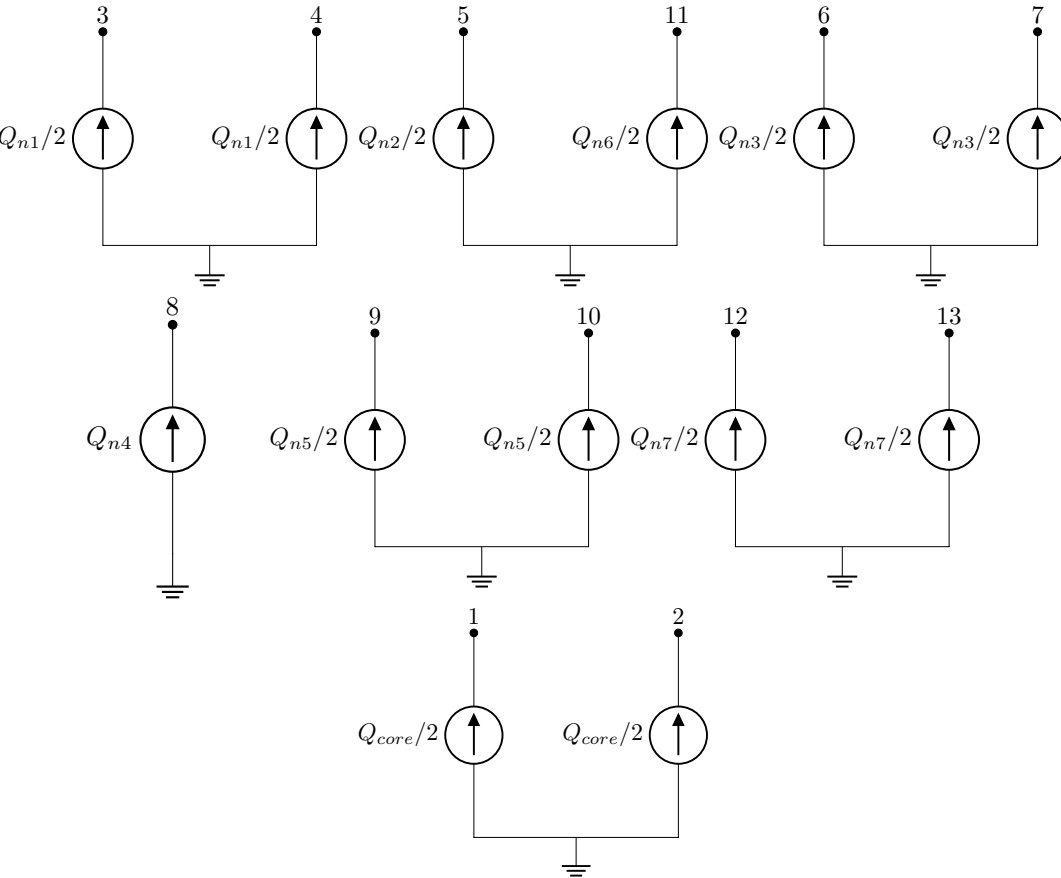

**Figure 6.** Heat injection schematic for the analyzed transformer (Section 3.2).

2.3.5. Complete Thermal Network of the Transformer

Finally, the previously described thermal sub-networks are assembled, obtaining the one shown in Figures 7 and 8.

*2.4. Simplified Thermal Network and Equation*

The thermal network obtained in Section 2.3 can be excessively complex to be solved without an electrical simulator. As a consequence, it needs to be simplified into an analytical equation that allows to obtain the temperature of the most representative nodes. A method to achieve that simplification systematically consists in using the Thevenin's theorem (Theorem 1) to obtain the equivalent circuit at that node.

**Theorem 1.** *Any linear circuit containing several voltages and resistances can be replaced by an equivalent circuit with just a single voltage source in series with a single resistance connected to a load. The equivalent voltage is the voltage obtained at the output terminals of the original circuit with those terminals open circuited. The equivalent resistance is the resistance that the original circuit would have between its output terminals if all independent sources were nullified.*

Applying Thevenin's theorem between the node in which the temperature should be measured and the earth of the circuit, it is possible to obtain an equivalent circuit in which that temperature corresponds to the open circuit voltage. Thus, the simplified equation that represents the temperature at that node is the parametrized Thevenin's voltage at that node.

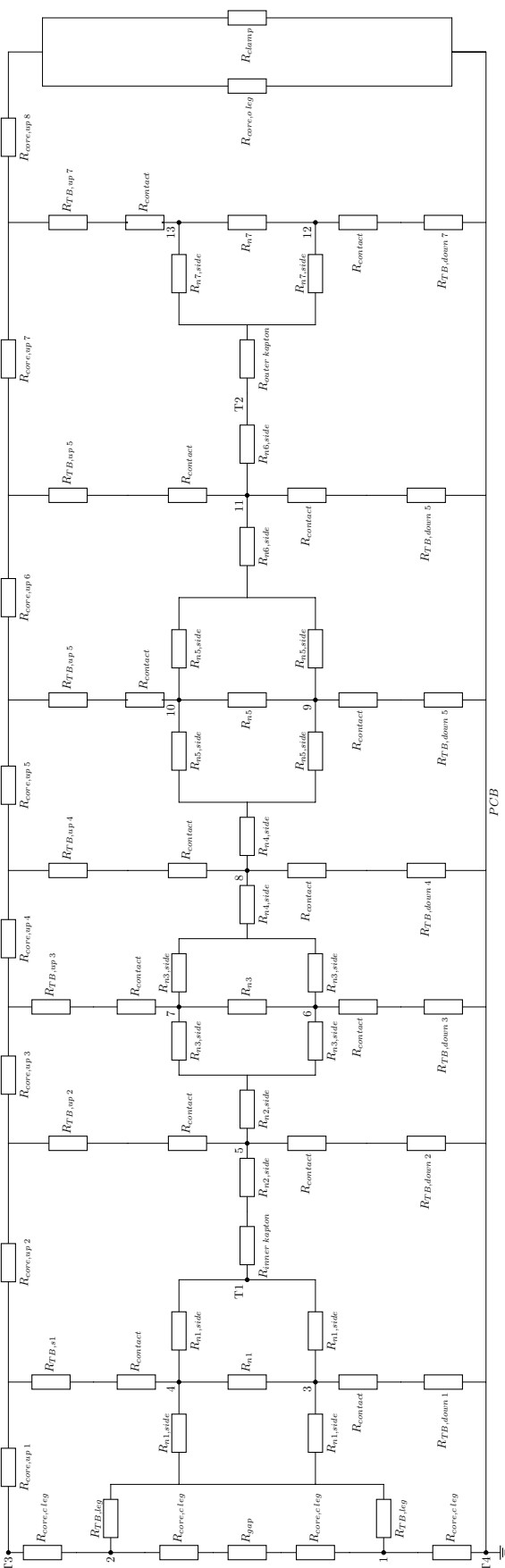

**Figure 7.** Complete thermal network of the analyzed transformer (Section 3.2): Main body.

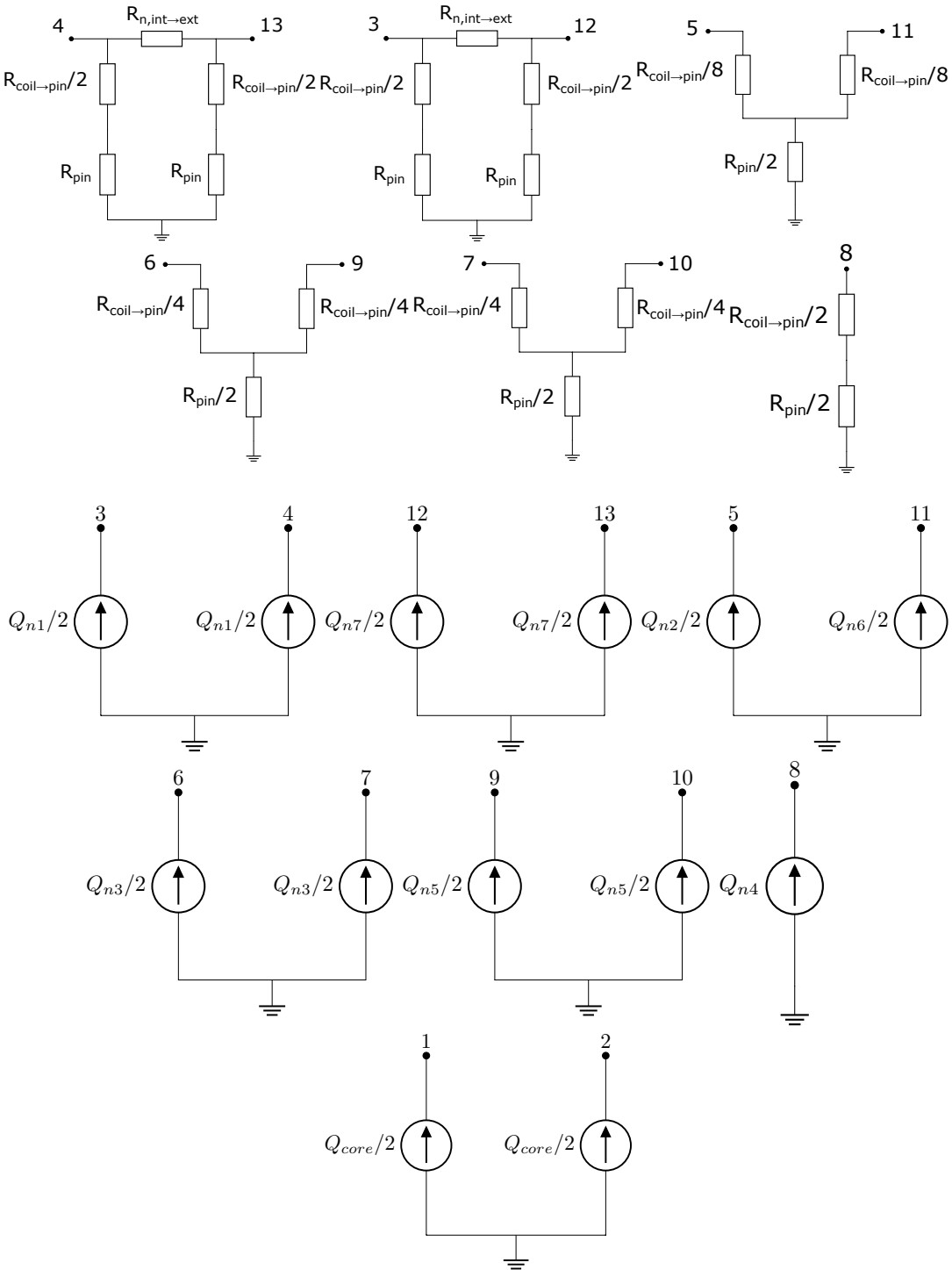

**Figure 8.** Complete thermal network of the analyzed transformer (Section 3.2): Connection to PCB and heat injection.

Due to the linearity of the circuit, it is possible to apply the superposition theorem (Theorem 2) to obtain the parametrized Thevenin's voltage as shown in Equation (18), using a circuit simulator to obtain the Thevenin's voltage for each element that produces heat separately.

$$U_{Thevenin} = \sum_i Q_i * U_{Thevenin} \left( Q_i = 1, \, Q_{j \neq i} = 0 \right) \tag{18}$$

where $U_{thevenin}$ is the Thenenin's voltage that models the produced heat and $Q_i$ and $Q_j$ are the power losses produced in each element of the transformer.

**Theorem 2.** *The response of a linear circuit caused by the simultaneous action of several independent sources is equal to the sum of the independent effects of each of them.*

$$f(c_1 * x_1, c_2 * x_2) = c_1 * f(x_1) + c_2 * f(x_2) \tag{19}$$

*where $f(x)$ is a linear function and $c_1$ and $c_2$ are constants.*

Using this method, the detailed thermal model was simplified into Equation (20).

$$\Delta T_{node} = c_1 * Q_{primary} + c_2 * Q_{secondary} + c_3 * Q_{core} \tag{20}$$

where $\Delta T_{node}$ is the temperature increment in a node of the detailed thermal network above the PCB temperature, $Q_{primary}$ is the heat produced in the primary winding of the transformer, $Q_{secondary}$ is the one produced in the secondary winding, $Q_{core}$ are the power losses produced in the core and $c1$, $c2$ and $c3$ are constants.

The simplified Equation (20) can be drawn as the circuit representation from Figure 9, which is much simpler than the original thermal network.

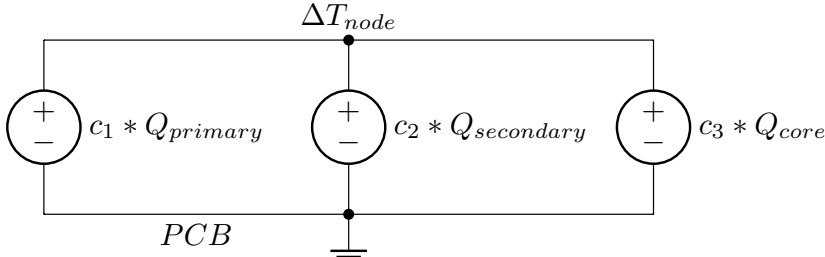

**Figure 9.** Thevenin's equivalent at a node of the detailed thermal network of the analyzed transformer (Section 3.2).

## 3. Experimental Validation

In this section, the accuracy of the developed thermal networks and the proposed simplified equation is analyzed and compared with experimental results.

### 3.1. Experimental Setup

In order to verify the analytical thermal model two different setups were used. The first one used a Radio-Frequency (RF) amplifier to inject a sinusoidal voltage waveform in the primary winding of the transformer (Figures 10 and 11). By means of this setup, core losses and winding losses can be measured to ensure that the calculated values, using Steinmetz equation for the core losses and the winding losses with the AC resistance of the windings obtained with a simulation of the transformer made with ANSYS Maxwell, are accurate (see Section 2.1 for more details). As a consequence of these power losses, certain temperature rise occurs in the transformer, which is measured and used to quantify the accuracy of the proposed method. The temperature measurement is explained later.

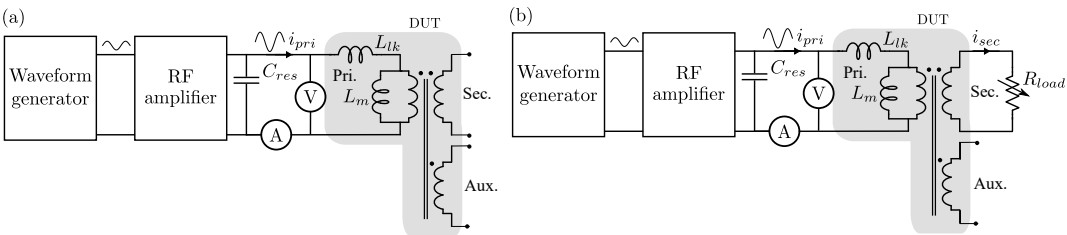

**Figure 10.** Electrical setup for tests using the RF setup: (**a**) With the secondary winding in open circuit and (**b**) with a load connected to it. The auxiliary winding remains an as open-circuit since its nominal load is negligible compared to the main load.

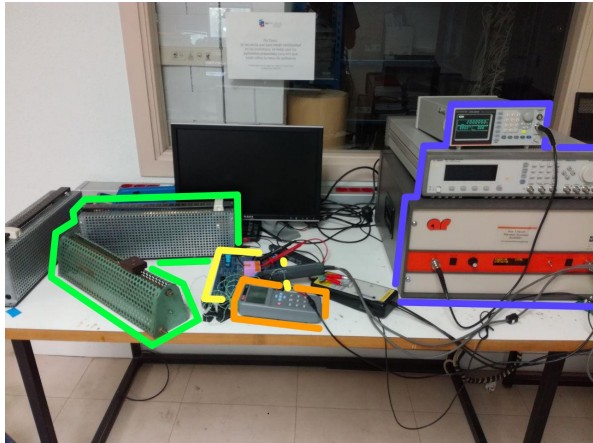

**Figure 11.** Photograph of the experimental setup for conducting the described RF test. In this setup, the input waveforms were generated using a signal generator and an RF amplifier (blue). The transformer and the required input capacitors were mounted in the PCB designed for both setups (yellow), and different loads (green) were used to measure the thermal behaviour at different operating points. During those tests the temperature measured by the thermocouples was registered using a datalogger (orange) and an oscilloscope was used for verifying the input and output waveforms (not shown in the photo).

The value of the resonant capacitor for this setup is marked in Table 5.

**Table 5.** Parameters used for the RF setup.

| Identifier | Value |
|:----------:|:-----:|
| $C_r$ | 11.3nF |

A second setup is proposed to prove the accuracy of the proposed method in a real operation of the prototype transformer in a power converter for space applications. These tests consists in using the transformer in a Flyback converter (Figures 12 and 13). These tests were carried out to ensure that the model was able to predict the reached temperature when non-sinusoidal currents were applied to the transformer. In this case, core losses were estimated using iGSE equation and winding losses applying the same AC resistance as the previous test to each of the harmonics of the current waveforms in each winding of the transformer (see Section 2.1 for more details).

To measure the temperatures of the transformer and PCB during those tests, several thermocouples were included between its windings and on its upper surface as well as bellow the transformer in contact with the PCB. The location of those sensors is shown in Figure 14. In particular, a four-channel RS PRO RS-1384 thermometer [39] with four type-K thermocouples [40] were used in this study to register the temperature rise of the magnetic component.

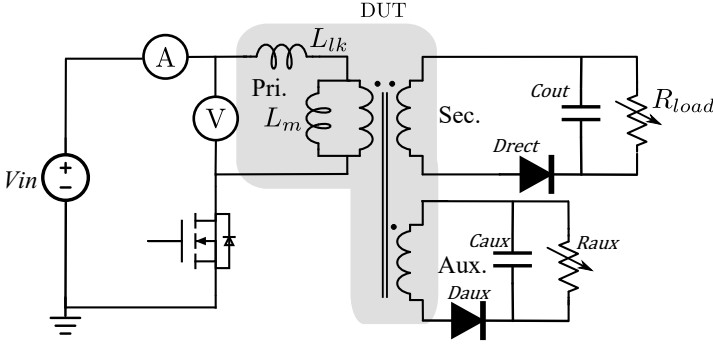

**Figure 12.** Electrical setup for tests using the Flyback converter setup.

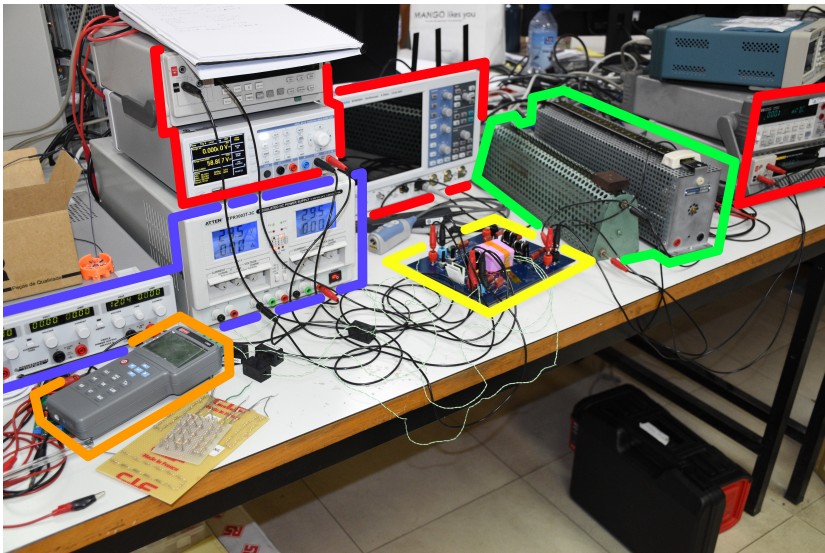

**Figure 13.** Photograph of the experimental setup for conducting the described Flyback test. In this photo, it is possible to discern the Flyback converter (yellow), the main voltage source and the auxiliar one that powers the control circuit (blue), the input and output voltage and current measurements (red), the loads used during the different tests (green) and the datalogger that registers the temperature measured by the thermocouples (orange).

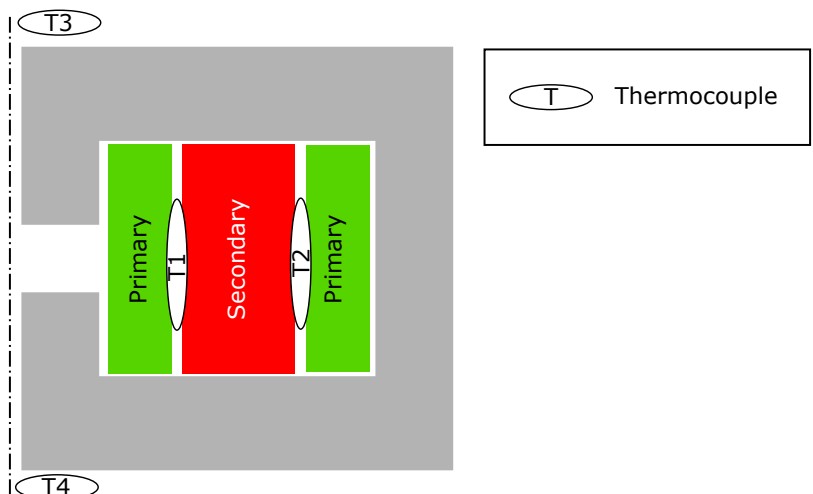

**Figure 14.** Location of thermocouples used for measuring the temperature during the conducted tests.

In both setups, heat was dissipated from the PCB through natural convection, as the PCB was supported by six legs to stabilize it. However, the final PCB will be mounted on a baseplate connected to a liquid cooling system. As a consequence, the temperature of that PCB is an input provided by the engineers that design the cooling system at Thales Alenia Space in Spain. Therefore, guaranteeing that the results of the test can be extrapolated to the final design is necessary. To achieve that, the PCB was used as a temperature reference point, allowing to infer the behaviour of the transformer in outer space from the conducted tests due to the linearity of thermal conduction. Nonetheless, that requires reducing the amount of heat dissipated from the transformer through convection and radiation to ensure that all the power losses generated in its windings and core were transmitted to the PCB though conduction. To achieve this, the transformer was thermally isolated by covering it with the insulating box shown in Figure 15. This allowed to conduct the tests without requiring a vacuum chamber that would have impeded measuring the electrical signals in the transformer.

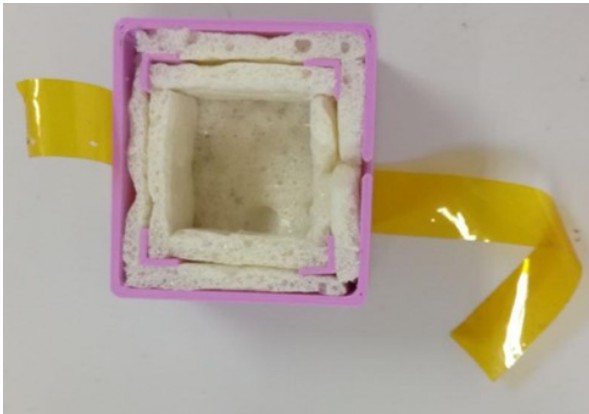

**Figure 15.** Insulating box used for reducing irradiation and convention during the conducted tests.

*3.2. Case Study*

The thermal network described in this document was developed for a transformer designed to operate in a Flyback converter for space applications, Figure 2 and Table 6. This converter was designed to be highly configurable for different applications, which is reflected in the use of several primary and secondary windings that can be connected in several configurations to achieve a high input and output voltage range of the converter. Therefore, tests were conducted for those configurations in which currents in both windings reached their maximum values as in those configurations the produced losses reach their maximum value. Nevertheless, the obtained thermal network is agnostic to the connections between those windings and, consequently, can be used for all configurations, see Table 6.

**Table 6.** Transformer characteristics: Core and number of conductor turns per layer.

| Identifier | Value | Identifier | Value | Identifier | Value | Identifier | Value |
|---|---|---|---|---|---|---|---|
| Core | RM8/I | Layer $N_2$ | 12 | Layer $N_4$ | 12 | Layer $N_6$ | 9 |
| Layer $N_1$ | 24 | Layer $N_3$ | 12 | Layer $N_5$ | 24 | | |

Regarding the design of the transformer, its windings were made from AWG 30 PTFE insulated conductors to optimize them due to the skin effect at the switching frequency of the converter. This frequency was also one of the reasons why the chosen core material was 3C95 ferrite, being the other the thermal stability of the component, see Table 7.

**Table 7.** Loss coefficients of 3C95 ferrite (table created by the authors, taken the values from [41]).

| Parameter | Value | Parameter | Value |
|---|---|---|---|
| Cm | $7.47 * 10^{-3}$ | Ct | 1.654 |
| Ct1 | $1.26 * 10^{-2}$ | Ct2 | $6.06 * 10^{-5}$ |
| $\alpha$ | 1.955 | $\beta$ | 3.07 |

Due to the need to reduce irradiation and convention effects during the tests to emulate the conditions in which the transformer will operate in space, an insulating box was designed to cover it while the tests where conducted. This box consists in a 3D printed box filled with insulating foam that fits in the available space around the transformer in the PCB to reduce the air layer around it (Figure 15).

However, the use of this element impedes the measurement with a thermal camera. As a consequence, several thermocouples were used to measure the temperature inside the transformer and inside the box while the tests were conducted, as well as the temperature of the PCB. Those thermocouples were verified before conducting the test using a thermal camera in order to detect any possible malfunction of those sensors (Figure 16).

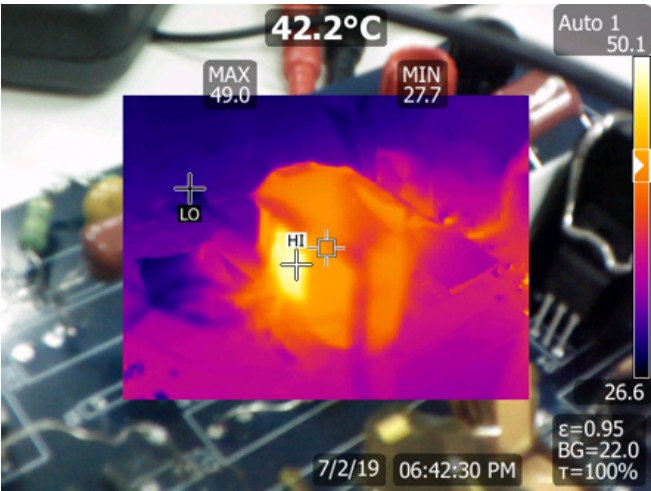

**Figure 16.** Conducted verification of thermocouple measures using a thermal camera.

### 3.3. Proposed Simplified Thermal Network and Equation

Using the method described in Section 2.4 the datailed thermal model shown in Figure 7 was simplified into the Thevenin's equivalent of that network at each relevant node and, in particular, at nodes in which a thermocouple were located. Comparing the equations obtained at the analyzed nodes it was possible to simplify all those equations into one that estimates the temperature increment of the windings over the temperature of the PCB and another one for estimating the increment on the top surface of the core, Equation (21), Figure 17.

$$\begin{cases} \Delta T_{windings} = 13 * Q_{primary} + 13 * Q_{secondary} + 5 * Q_{core} \\ \Delta T_{core} = 6.5 * Q_{primary} + 6.5 * Q_{secondary} + 13 * Q_{core} \end{cases} \tag{21}$$

where $\Delta T_{node}$ is the temperature increment in a node of the detailed thermal network above the PCB temperature, $Q_{primary}$ is the heat produced in the primary winding of the transformer, $Q_{secondary}$ is the one produced in the secondary winding and $Q_{core}$ are the power losses produced in the core.

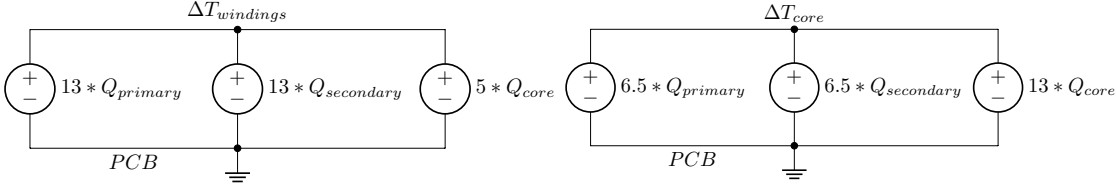

**Figure 17.** Simplified thermal network of the transformer.

### 3.4. Results Comparison

The verification of the proposed thermal network was achieved through a series of experiments using the setups described in Section 3.1 in which the steady-state temperature increment in each thermocoulple over the PCB, registered after the thermal balance in the thermocouples was rached, was measured under different electrical configurations. Those temperatures are the ones required for verifying the fulfillment of the ECSS standards and, therefore, need to be used during the design of the transformer and converter.

The electrical configuration of each of those experiments can be found in Table 8 whereas the reached steady-state temperature increment over the PCB in each thermocouple are shown in Figures 18–20 (labeled as 'Experimental results' in the legend), being the sample number the number of the experiment in which those values where recorded.Those figures also show the estimated

temperature increment using the detailed thermal network (Figure 7) labeled as 'Detailed thermal network ', the proposed simplified network (Figure 17, whose behaviour is characterized by the Equation (21), thus the label 'Simplified Equation' in the legend) and the least-squares best-fit line obtained with the Regression Learner Matlab Toolbox, labeled as 'Linear regression'. This linear regression is made using the experimental measurements to obtain the coefficients $c_1$, $c_2$ and $c_3$ from Equation (20). Both models, the proposed thermal network and the least-squares best-fit line, required estimating the heat produced in the core and windings of the transformer. This was achieved using the method explaned in Appendix A, being shown the estimated values in Table 9. In these results, it is possible to discern that the simplified network (and its associated simplified equation obtained analytically) has a similar precision to the one achieved by a linear regression without requiring to conduct any test to obtain it. Moreover, the proposed model is able to predict with a reasonable precision the temperature increment in the worst case, allowing to verify if the transformer will be suited for its intended application, which is an important stage of the converter design process when a configurable transformer is used.

**Table 8.** Electrical configuration in each of the conducted experiments and measured absolute temperature in T4 thermocouple after reaching the thermal steady state.

| Experiment | Setup | Fundamental Frequency (kHz) | $V_{in,RMS}$ (V) | $I_{in,RMS}$ (A) | $V_{out,RMS}$ (V) | $I_{out,RMS}$ (A) | T4 (°C) |
|---|---|---|---|---|---|---|---|
| 1 | Flyback | 200 | 17.41 | 2.08 | 8.40 | 3.34 | 42.8 |
| 2 | Flyback | 200 | 58.55 | 0.59 | 8.36 | 3.31 | 37.7 |
| 3 | Flyback | 200 | 58.71 | 0.28 | 8.14 | 1.56 | 32.8 |
| 4 | Flyback | 200 | 17.86 | 0.90 | 8.27 | 1.53 | 33.2 |
| 5 | Flyback | 200 | 58.86 | 0.28 | 8.15 | 1.55 | 33.3 |
| 6 | Flyback | 200 | 58.80 | 0.27 | 4.02 | 3.1 | 32.1 |
| 7 | Flyback | 200 | 17.71 | 1.10 | 4.02 | 3.67 | 33.2 |
| 8 | Flyback | 200 | 17.26 | 2.37 | 4.09 | 6.88 | 55.1 |
| 9 | Flyback | 200 | 58.71 | 0.58 | 3.98 | 6.59 | 39.6 |
| 10 | RF | 200 | 33.21 | 2.43 | 8.56 | 7.35 | 29.7 |
| 11 | RF | 400 | 66.01 | 2.72 | 16.37 | 8.39 | 32.3 |
| 12 | RF | 600 | 64.29 | 1.86 | 16.57 | 5.90 | 31.6 |
| 13 | RF | 200 | 63.74 | 1.11 | 17.06 | Open circuit | 27.9 |
| 14 | RF | 400 | 63.81 | 0.54 | 17.19 | Open circuit | 26.4 |
| 15 | RF | 600 | 63.25 | 0.33 | 17.06 | Open circuit | 26.3 |
| 16 | RF | 200 | 17.19 | 1.26 | 4.18 | 4.11 | 26.2 |
| 17 | RF | 400 | 34.39 | 1.41 | 8.98 | 4.56 | 27.2 |
| 18 | RF | 600 | 36.39 | 1.05 | 9.39 | 3.45 | 26.6 |
| 19 | RF | 200 | 39.36 | 0.65 | 10.91 | Open circuit | 25.5 |
| 20 | RF | 400 | 50.69 | 0.42 | 13.67 | Open circuit | 25.4 |
| 21 | RF | 600 | 51.10 | 0.27 | 13.88 | Open circuit | 24.9 |

**Table 9.** Thermal parameters used for estimating the temperature increment with the proposed model.

| Experiment | Setup | $Q_{core}$ (mW) | $Q_{primary}$ (W) | $Q_{secondary}$ (W) |
|---|---|---|---|---|
| 1 | Flyback | 9.45 | 1.14 | 0.65 |
| 2 | Flyback | 67.20 | 0.28 | 0.30 |
| 3 | Flyback | 38.39 | 0.11 | 0.12 |
| 4 | Flyback | 10.61 | 0.28 | 0.18 |
| 5 | Flyback | 38.39 | 0.11 | 0.12 |
| 6 | Flyback | 20.41 | 0.12 | 0.17 |
| 7 | Flyback | 4.40 | 0.35 | 0.29 |
| 8 | Flyback | 3.77 | 1.80 | 1.38 |
| 9 | Flyback | 21.83 | 0.44 | 0.60 |
| 10 | RF | 5.56 | 0.19 | 0.17 |
| 11 | RF | 23.01 | 0.63 | 0.48 |

**Table 9.** *Cont.*

| Experiment | Setup | $Q_{core}$ (mW) | $Q_{primary}$ (W) | $Q_{secondary}$ (W) |
|:---:|:---:|:---:|:---:|:---:|
| 12 | RF | 13.60 | 0.52 | 0.35 |
| 13 | RF | 46.57 | 0.04 | 0 |
| 14 | RF | 21.90 | 0.02 | 0 |
| 15 | RF | 13.04 | 0.02 | 0 |
| 16 | RF | 0.75 | 0.05 | 0.04 |
| 17 | RF | 3.16 | 0.17 | 0.13 |
| 18 | RF | 2.39 | 0.16 | 0.11 |
| 19 | RF | 10.66 | 0.01 | 0 |
| 20 | RF | 10.72 | 0.01 | 0 |
| 21 | RF | 6.92 | 0.01 | 0 |

As to the details of the measurements shown in Figures 18–20, it is possible to discern that the registered temperatures in the thermocouples located inside the windings (Figures 18 and 19) are similar as the model predicted. This is a consequence of the much higher thermal conductivity of copper compared to the one of the rest of the materials used in the transformer and the small thickness of the resin that covers the conductors and the one of the kapton layers.On the other hand, temperatures reached in the core (Figure 20) lower than the ones measured in the windings, which was expected as the main power losses produced in the transformer are the ones disipated by the windings. These losses are then conducted to the PCB through the core due to the lack of convection and radiation, heating this element. As a consequence of the smaller thermal conductivity of 3C95 ferrite, the temperature gradient in this element is larger than the one in the rest of the transformer, which explains the difference between the temperature of T3 and the one registered by the thermocouples inside the windings.

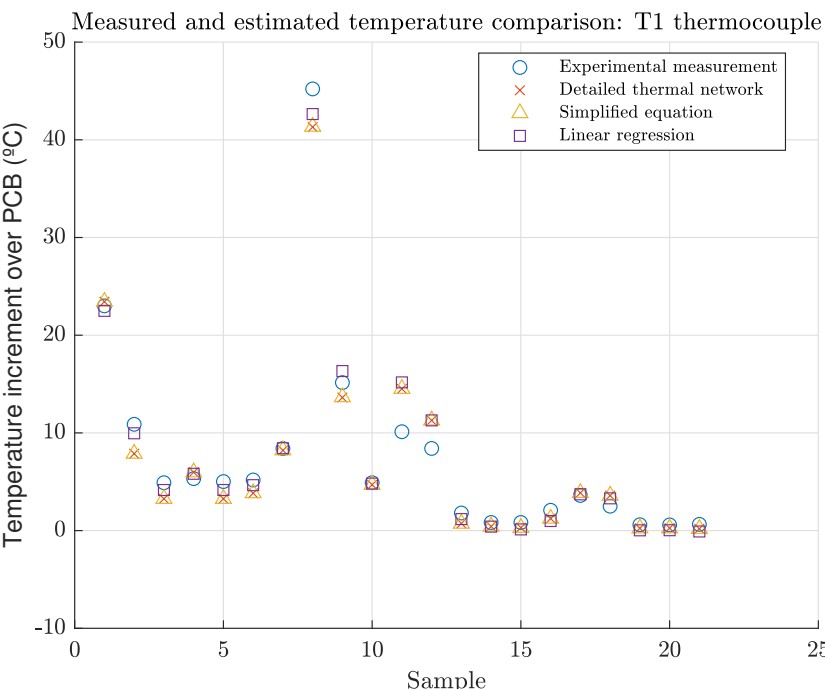

**Figure 18.** Comparison of measured and estimated temperature increment in T1 thermocouple over the temperature reference point. This thermocouple is located at the interface between the inner part of the primary winding and secondary (see Figure 14).

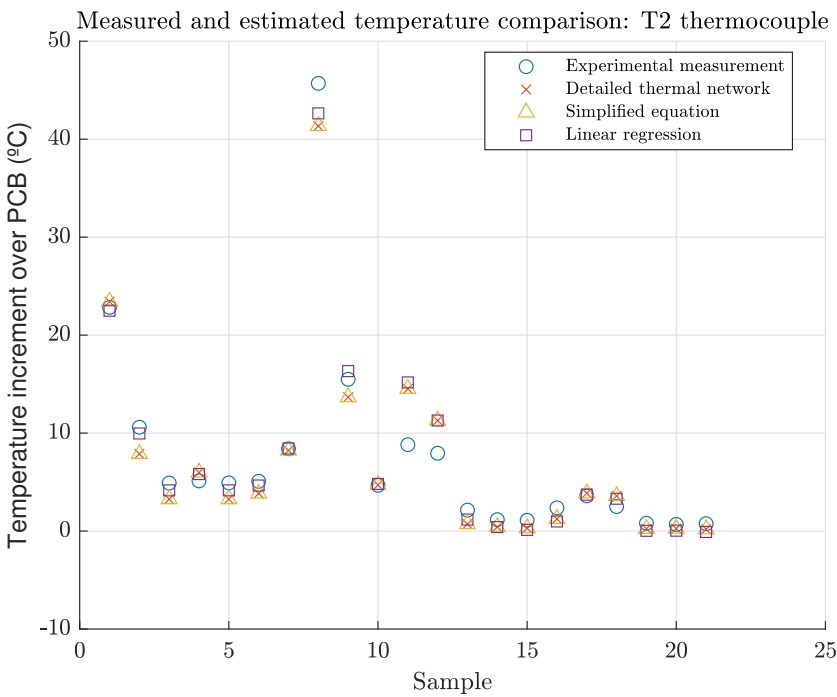

**Figure 19.** Comparison of measured and estimated temperature increment in T2 thermocouple over the temperature reference point. This thermocouple is located at the interface between the outer part of the secondary winding and the outer primary winding (see Figure 14).

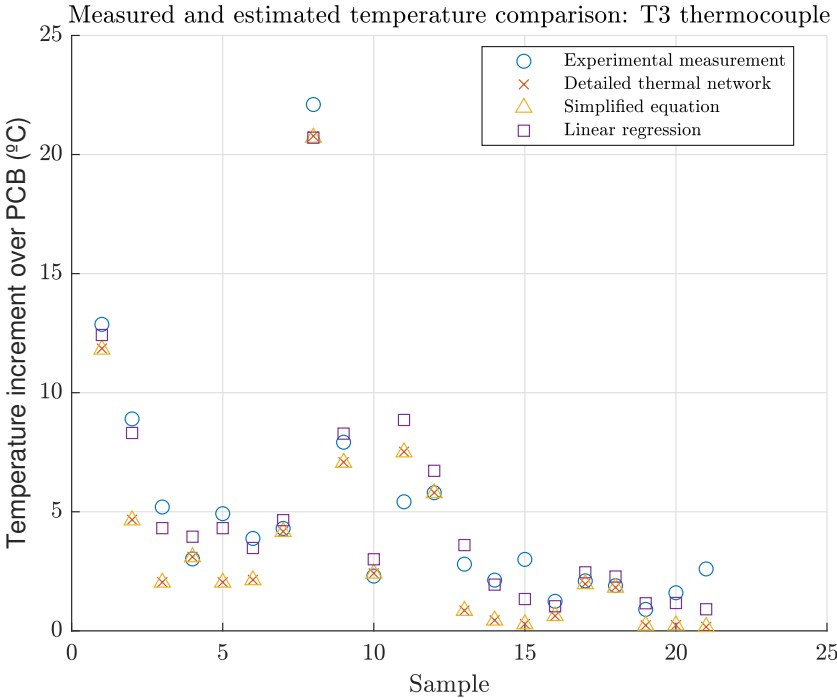

**Figure 20.** Comparison of measured and estimated temperature increment in T3 thermocouple over the temperature reference point, corresponding to the hotspot temperature of the ferrite core, located on its top surface (see Figure 14).

On the other hand, it is possible to appreciate that the proposed model accurately predicts the temperature in the most critical configuration (Experiment 8 of Table 8) in all thermocouples. This is important as that temperature is the one that limits the operation range of the converter. Consequently, having a model that correctly predicts it, allows to evaluate if the converter will be able to operate at

a given conditions that might not have been considered in the specifications without damaging the transformer. In this case, the applied standards are: [2–6], which, in conjunction, establish a maximum allowed temperature of 50 °C over the PCB temperature (temperature reference point). This condition can be accurately checked by means of the proposed model, as shown in the results from this section.

Definitively, it can be stated that the accuracy of the proposed thermal network and its simplified is within the band of 1 °C to 5 °C compared to measurements, which is quite acceptable for these power electronic applications, as found in the carried-out literature review. The suggested methodology allows power electronic designers to properly characterize the thermal behaviour of their magnetic components and ensure that they comply with the required specifications, by means of the proposed simple, accurate and very low time-consuming thermal equation (Equation (20)), which is the main contribution of this paper.

## 4. Conclusions

A systematic methodology to obtain an accurate thermal model based on thermal networks for magnetic components in space applications is introduced in this paper. It can be used for any inductor or transformer from any power system in space environmental conditions. In order to analyze the wide scope of this methodology, a generic Flyback transformer with multiple layers is used in this study, to cope with a wide range of common details and possibilities.

Furthermore, a simplification of the network using the Thevenin's theorem is proposed, so that the result is a very simplified circuit. This circuit is characterized by a simple equation, which is also presented. This methodology allows power electronics designers to properly analyze the thermal behaviour of magnetic components in a simple way, avoiding the extensive time consumption of other approaches like the use of FEA tools or the creating of physicial prototypes, while obtaining accurate results.

In order to validate the proposed methodology, two experimental setups are proposed, based in the use of an RF amplifier and using a power converter, respectively. Comparing the measurements of the tested custom-made Flyback transformer with the suggested models, deviations of only 1 °C and 5 °C are achieved, which is highly acceptable for these applications. Definitely, the proposed approach is proven valid and accurate.

**Author Contributions:** Conceptualization, G.S. and D.d.l.H.; methodology, G.S.; software, D.d.l.H.; validation, G.S., D.d.l.H. and V.Š.; formal analysis, V.Š. and P.A.; investigation, G.S., D.d.l.H., V.Š. and P.A.; resources, V.Š. and P.A.; data curation, G.S. and V.Š.; writing–original draft preparation, G.S. and D.d.l.H.; writing–review and editing, G.S., D.d.l.H., V.Š. and P.A.; visualization, G.S. and D.d.l.H.; supervision, V.Š. and P.A.; project administration, V.Š. and P.A.; funding acquisition, V.Š. and P.A. All authors have read and agreed to the published version of the manuscript.

**Funding:** David De-La-Hoz holds a grant for master students by Thales-Alenia Space through its participation in the program 'Industrial Council', organized by the 'Centro de Electrónica Industrial' (http://www.cei.upm.es/cei-industrial-council/). Guillermo Salinas holds a predoctoral contract under RD99/2011, funded by the 'Programa Propio' grant from the 'Universidad Politécnica de Madrid' (Project N070505C709).

**Conflicts of Interest:** The authors declare no conflict of interest. The funders had no role in the design of the study; in the collection, analyses, or interpretation of data; in the writing of the manuscript, or in the decision to publish the results.

## Abbreviations

The following abbreviations are used in this manuscript:

| | |
|---|---|
| FEA | Finite Element Analysis |
| PCB | Printed Circuit Board |
| PCDU | Power Control and Distribution Unit |
| PTFE | Polytetrafluoroetheylene |
| RF | Radio-Frequency |
| TB | transformer bobbin |

## Appendix A. Power Loss Estimation

The proposed thermal network is independent of the used methodology for estimating the losses in the windings and the core of the transformer. Nevertheless, faster analytical methods like Ferreira equations or lumped circuit models of the transformer are also compatible. Because of that, this appendix should be understood as a description of the process followed by the authors to validate experimentally the proposed model and as a guide to estimate those losses from the obtained impedance model from a finite element analysis simulation.

*Appendix A.1. Core Losses Estimation*

The RF setup allows to verify the behaviour of the transformer with sinusoidal waveforms. Hence, it is possible to estimate the core losses in this setup using the Steinmetz equation (Equation (A1)) using the parameters provided by its manufacturer (Table 7). In this setup, it is possible to obtain the peak value of the magnetic field using the Faraday-Lenz law (Equation (A3)) as the input voltage of the transformer can be measured.

$$P_v = k * f^\alpha * B^\beta \tag{A1}$$

where $P_v$ is the time average core power loss per unit volume, f is the frequency of the sinusoidal waveforms of both the voltages and the currents in the transformer, B is the peak value of the magnetic field and k, $\alpha$ and $\beta$ are the coefficients provided by the manufacturer to adjust the equation to a given material. The Steinmetz coefficient k is temperature dependant, so in order to estimate the losses, it was obtained for the worst case temperature (100 °C ) using Equation (A2) [41]

$$k(T) = Cm * \left( Ct2 * T^2 - Ct1 * T + Ct \right) / 1000 \tag{A2}$$

where Cm, Ct, Ct1 and Ct2 are coefficients provided by the manufacturer for a particular material (Table 7)

$$= -N * A_e \frac{dB}{dt} \tag{A3}$$

where $V$ is the induced voltage in a winding, $A_e$ is the specific area of the core, $N$ is the number of turns of the winding and $B$ is the magnetic field.

Contrarily, it is not possible to use Steinmetz equation for estimating the losses due to their non-linear nature. Instead, the IGSE equation (Equation (A4)) needs to be used, as it provides better results with the same parameters as the previous one.

$$P_v = \frac{1}{T} \int_0^T k_i \left| \frac{dB}{dt} \right|^\alpha \left( \Delta B^{\beta-\alpha} \right) dt \tag{A4}$$

where $P_v$ are the time average power losses per unit volume, T is the period of the magnetic field waveform, B is the magnetic field, $k_i$ is a parameter that can be obtained with Equation (A5) and $\alpha$ and $\beta$ are the Steinmetz parameters of the core material.

$$k_i = \frac{k}{(2\pi)^{\alpha-1} \int_0^{2\pi} |cos\theta|^\alpha \, 2^{\beta-\alpha} d\theta} \tag{A5}$$

where k (Equation (A2)), *alpha* and *beta* are the Steinmetz parameters of the material. In order to estimate the core power losses in Flyback setup, k was obtained at the worst case temperature (100 °C).

*Appendix A.2. Winding Losses*

The resistance of a transformer winding is frequency dependant mainly due to the skin, proximity and gap effects. In order to take them in account, this resistance was extracted from a finite element simulation conducted with ANSYS Maxwell. However, the resistivity of a conductor is temperature

dependent and, therefore, it also needs to be taken into account. To solve this, the transformer was simulated at 100 °C as it was the worst-case temperature measured in the tests.

Once the impedance model of the transformer was obtained using ANSYS, it was possible to obtain the loss contribution of each harmonic of the measured current waveforms in the windings using Ohms law (Equation (A6)). Nevertheless, it needs to be taken into account that the Fourier transform of the Flyback current waveforms is a sum of infinite frequency harmonics. However, the winding losses produced in those tests can be characterized by the first ten harmonics of the signal without sacrificing the accuracy of the results.

$$P_{wind} = R_{DC} * I_{DC}^2 + \sum_{harmonic} \left( \sum_{i=wire} \left( \sum_{j=wire} R_{ac_{ij}} \cdot I_{rms_i} * I_{rms_j} \right) \right) \tag{A6}$$

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
