# Peer review of "Simplification of Thermal Networks for Magnetic Components in Space Power Electronics"

_energies, doi:10.3390/en13112903_

Round 1
Reviewer 1 Report
The authors present their work on thermal analysis of a transformer for space power electronics applications. The research focus has been placed on developing an accurate thermal modelling approach. Here, an equivalent circuit approach has been employed. A model reduction technique, standard circuit theory, has been used to reduce the model resolution to a required form. Both theoretical and experimental results are presented showing good correlation.
The problem I am having with the paper is that the research objectives and findings are not well explained. Although, the authors provide some detailed information on selected aspects of their work, the overall picture is rather vague, in my opinion. A list below provides detailed comments and suggestions for the authors’ consideration. This is to aid revision process of the existing manuscript.
- It is unclear why the authors selected the specific thermal modelling route, i.e. there are alternative methods allowing for accurate low-order thermal lumped-parameter equivalent-circuit model representation. Please, see the following papers, R. Wrobel, et al., “A General Cuboidal Element for Three-Dimensional Thermal Modelling,” IEEE TM, 2010; N. Simpson, et al. “Estimation of Equivalent Thermal Parameters of Impregnated Electrical Windings,” IEEE TIA 2013;
- It would be good to introduce a specification for the analysed converter, including environmental factors and operating regime for the developed inverter. The authors should also articulate a bit better the need for thermal analysis of the transformer, (wound magnetic component) on its own;
- The level of detail on the transformer construction and materials used is insufficient to make a proper judgment on formulation of the thermal model used. Is the transformer, winding body impregnated? What is the type of winding/conductor used? What is the conductor and insulator grade? What is the core material? What is the specific power loss data for the core material used? Etc.;
- It is unclear how the authors calculated and measured the individual power loss components, i.e. winding and core power losses;
- The experimental procedure is unclear. A more detailed description of the test setup and testing procedure would be useful here. E.g. how the generated heat is removed from the transformer body via the PCB assembly?
- There are numerous contact thermal resistances in the analysed transformer assembly. It is unclear how the authors informed these parameters and then accounted for them in the thermal model;
- Both the winding and core power losses are temperature dependent. How did the authors account for these in the thermal analysis?
- The nomenclature used in the paper is confusing, e.g. thermal resistance is given in Kelvin/Watt or Celsius/Watt not in Ohm. The authors should carefully separate and define the electrical and thermal quantities.
- The authors state that the heat transfer due to convection and radiation can be neglected in the analysis. To avoid confusion, it might be worth clarifying that radiation is the main heat removal mechanism for space applications. However, heat transport from the heat source to the heat sink is frequently assured by other heat transfer mechanism including conduction and convection, e.g. thermal management systems with phase change;
- Literature references section needs to be expend and updated to account for the existing state of the art in thermal modelling, design of passive wound components.
My recommendation for the authors is to take a step back and rethink the main focal point of the paper. At the moment neither of the inverter, transformer development nor the proposed thermal modelling approach are well presented (with sufficient level of detail).
Author Response
Dear Reviewer,
Please, find an answer point-by-point attached as a PDF file. The modifications in the paper will be shown in red font.
Thank you for your time,
The Authors

Reviewer 2 Report
The manuscript presents an interesting research topic.
In my opinion, the major revision is required according to the following suggestions:
- The introduction section should highlight what is the novelty of the presented research.
- The literature survey could be extended on the positions about the application of thermal models based on thermal networks in space application.
- The introduction section does not contain the literature survey of the different methodology of the thermal modelling methods. Is thermal network the only one method of thermal modelling? If not, why this one was chosen and others were rejected?
- In line 16, the sentence is introduced: “Some regulations regarding the thermal limits…” . Could Authors present directly the limitations for investigated application? Does the developed device fulfil the criteria introduced by mentioned regulations? Is the analysed device designed to work in outer space conditions or on-board internal environment where people spend time? That information is not presented directly and assumed boundary conditions and assumptions should be fitted to the destination location.
- In lines 72-90, Authors mentioned about the FEM model used to calculate heat sources used next in the thermal networks. However, the developed FEM model description was not introduced. The values of the power losses were not introduced also. In the manuscript, Qcore and Qwindings symbols describing power losses in the specific components were presented without any values. It is commonly known that for example conductor resistance is dependent on temperature and it influences the power losses. In outer space, the temperatures could be varied in a very wide range (e. g. depending on sunlight exposition). How did Authors deal with this problem? In my opinion, the power losses description should be extended and additional explanation about the influence of the external conditions could be added.
- In line 103, Authors mentioned about PCB where the power losses are dissipated. What is the temperature range of this equipment? This temperature is introduced as the constant. This remark is connected with the previous point. The temperature of the PCB which was measured during the experimental part by thermometer no. 4 could be introduced.
- In my opinion, the lines 129-133 are not clear and should be rewritten to clarify the authors' minds. The windings in the transformers and motors are characterised by anisotropic thermal conductivities. Considering the axis of the coils wounding the axial and radial thermal conductivity is relatively small (closer to the thermal conductivity of the insolation). At the same time, the tangential thermal conductivity is relatively big (close to the thermal conductivity of the copper). Was it considered within the presented research? If yes, please point where is this description in the answer.
- What is the thermal conductivity presented in line 160? Is it copper or the conductivity covering the resin lamination of each coil? It could be presented directly.
- In Tab. 2, for the first time symbol cw appeared. What does it mean?
- In line 328, Authors described temperature sensors. There is no such as fiber-optics thermocouples. The thermocouple is a device based on the Seebeck effect. The better description could be fiber-optics temperature sensor or thermometer.
- Please, explain why the temperature T1, and T2 present almost the same value. It should be also comment in the conclusion section.
- The results subsection should highlight the novelty of the research.
Editorial remarks:
- In Fig. 1, there are missing pointers (a) and (b).
- In the description of Fig. 2, the capital letter in word “Corresponding” occurs. In my opinion, the photo of the investigated device could be helpful in this place.
- In Fig. 2, the first time the coil winder word is used and it is repeated many times in manuscript (sometimes as coil winder). Is it the dedicated word to this component description? Coilwinder is a commonly used word to a machine which wounds windings. Did you mean the transformer bobbin? If yes, I could suggest changing this word in the whole manuscript.
- Few sentences, such as: “Nevertheless, the direct analytical solution of this equation is unfeasible in such a heterogeneous object as the magnetic components are, also due to their complex geometry.”, “The chosen domains for achieving that simplification were the windings, that can be considered as stacked torus, and the core, which was analyzed as a set of connected prisms.”, seem to be an unnecessary complex. They could be divided into separate sentences. Then, I would be more readable.
- Please, consider using an asterisk (*) as the symbol of multiplication. Is it recommended one by the journal? In Tab. 1, Authors used dots as the multiplication symbol.
- Why Authors introduced the symbol descriptions under each equation, beginning with a new paragraph and capital letter?
- In line 204, the introduced sentence could be placed directly behind the previous one. Corrected grammar should be “in Table 2.”.
- Please, consider moving some of the graphs of the thermal network to the appendix. The font size in Fig. 3 and Fig. 4 varies in high range.
Author Response
Dear Reviewer,
Please, find an answer point-by-point attached as a PDF file.
Thank you for your time,
The Authors

Round 2
Reviewer 1 Report
The authors significantly improved the manuscript, but the main issue remains. In my opinion, the presented model reduction using fundamental circuit theory is not new. As this aspect has been chosen by the authors as a focal point the paper, it is rather difficult to justify the work. Design considerations of magnetic components for space applications seem more relevant as a research subject worth a journal publication. The presented work is interesting and technically sound, but in my opinion, it should be refocused, to make it stand out.
A couple of minor comments to the revised text,
- Note that finite element approach used in thermal analysis, which the authors refer to in the paper allows for a simple analysis with any inputs. ‘…the aim of this paper is obtaining a thermal model, not a single solution of the heat transfer problem for a certain set of inputs, so that the temperature rise of the studied magnetic component can be accurately obtained for any input conditions (i.e. different core and winding power losses).’;
- It is not clear how the authors derived contact thermal resistance in a reliable manner. These usually require some form of experimental calibration. ‘…first approach requires prototype measurements, what is not always possible and is time consuming; we tried to avoid this situation to create a model that can be use in a design stage without the need of measurements nor prototypes.’;
- It would be useful to add some labels to the photographs illustrating experimental setups.
Author Response
Dear Reviewer,
Thank you very much for reviewing our manuscript and for your suggestions. Please, find attached our answer to your questions.
The Authors

Reviewer 2 Report
Thank you for the corrections introduced in the manuscript and comprehensive answers to my questions. In my opinion, the paper quality has been improved. I wish you good look with further researches and publications.
Author Response
Dear Reviewer
Thank you for reviewing our manuscript and for your encouraging words. Please, you can also find the new minor changes suggested by the Editor and the other reviewer updated in blue font, in the new revised manuscript.
The Authors